# Adaptation and Psychometric Validation of the EMPOWER-SUSTAIN Usability Questionnaire (E-SUQ) among Patients with Metabolic Syndrome in Primary Care

**DOI:** 10.3390/ijerph18179405

**Published:** 2021-09-06

**Authors:** Nurul Hidayatullaila Sahar, Siti Fatimah Badlishah-Sham, Anis Safura Ramli

**Affiliations:** 1Department of Primary Care Medicine, Faculty of Medicine, Universiti Teknologi MARA, Jalan Prima Selayang 7, Batu Caves 68100, Selangor, Malaysia; yuyulaar@gmail.com (N.H.S.); sfatimah31@uitm.edu.my (S.F.B.-S.); 2Institute of Pathology, Laboratory & Forensic Medicine (I-PPerForM), Universiti Teknologi MARA, Jalan Hospital, Sungai Buloh 47000, Selangor, Malaysia

**Keywords:** chronic care model, self-management, usability, questionnaire validation, Malay language

## Abstract

Self-management support is one of the most important components of the Chronic Care Model (CCM). The EMPOWER-SUSTAIN Global Cardiovascular Risks Self-Management Booklet^©^ was developed for patients with Metabolic Syndrome (MetS), inspired by the CCM. Assessing usability of a self-management tool is important in chronic disease management. However, there was no available instrument to assess usability of a self-management booklet, as most instruments were developed to assess usability of mobile application. Therefore, this study aimed to adapt *Skala Kebolehgunaan Aplikasi Mudah Alih* (SKAMA) into the EMPOWER-SUSTAIN Usability Questionnaire (E-SUQ) and to determine its validity and reliability in assessing usability of a self-management booklet. A cross-sectional validation study was conducted among patients with MetS attending a university primary care clinic in Selangor, Malaysia. Content validation, adaptation and face validation of E-SUQ were performed according to recommended guidelines. It underwent two rounds of content validation as major revision was required for item 5. Subsequently, the revised E-SUQ was face-validated by 10 participants. Psychometric evaluation was conducted using principal component analysis with varimax rotation to determine the underlying structure of E-SUQ. Internal consistency reliability was assessed using Cronbach’s α coefficient and the test-retest reliability was assessed using intraclass correlation coefficient (ICC (2,k)). A total of 205 patients participated in the study. The item-level content-validity-index (I-CVI) for item 5 improved from 0.57 to 1.0 after the second round of content validation. The final S-CVI/Ave value for ESUQ was >0.90. The item-level face-validity-index (I-FVI) ranged between 0.9 and 1.0. Kaiser-Meyer-Olkin value of 0.871 and Bartlett’s test of sphericity *p*-value of <0.05 indicated good sample adequacy for factor analysis. Two factors with eigenvalues of >1 were extracted according to the Kaiser’s Criteria. The two extracted factors explained 60.6% of the cumulative percentage of variance. The elbow of the scree plot occurred between the second and third component, suggesting two factors to be retained. The two factors were consistent with “Positive” and “Negative” tone model. The overall Cronbach’s α coefficient was 0.77, indicating good internal reliability. The overall ICC was 0.85, indicating good reproducibility. The E-SUQ is shown to be valid, reliable and stable to measure the usability of a self-management booklet among patients with MetS in a university primary care clinic in Malaysia.

## 1. Introduction

The increasing prevalence of cardiovascular (CV) risk factors which include Metabolic Syndrome (MetS) represents a public health issue of growing concern [1]. MetS is defined by a clustering of CV risk factors which include central obesity, dyslipidaemia, elevated blood glucose and elevated blood pressure [2]. Individuals with MetS were twice as likely to develop CV events, namely, heart attack and stroke [3]. The prevalence of MetS globally has reached epidemic proportion [4]. In Malaysia, 43.4% of Malaysian adults have MetS [5]. The Chronic Care Model (CCM) is a guide to transform the management of chronic conditions including MetS to improve outcomes in primary care [6]. It is a conceptual model for restructuring the healthcare system to ensure productive interactions between an informed, activated patient and a proactive, prepared practice team [7]. The CCM focuses on optimizing 6 interrelated key elements which include healthcare organization, delivery system design, patient self-management support, clinical information system, decision support and use of community resources [8].

Self-management support has been recognized as one of the most important components of the CCM [9]. Evidence has shown that self-management support by healthcare professionals and utilization of self-management tools were effective in improving knowledge, motivation, self-management behaviours and health outcomes in patients with MetS [10,11,12,13,14]. In Malaysia, a self-management support tool named the EMPOWER-PAR Global CV Risks Self-Management Booklet^©^ was developed in the EMPOWER-PAR study [15,16]. This booklet has recently been revised to become the EMPOWER-SUSTAIN Global CV Risks Self-Management Booklet^©^. It is given to patients so that they can be empowered to actively participate in the management of their own health. Therefore, it is important to assess the usability of this self-management booklet among the target population to ensure its usefulness.

Usability is defined as “the extent to which a product can be used by a specific user for a specific goal in a specific context or environment, and provides a satisfying experience” [17]. Two types of study that can be used to assess usability are qualitative or quantitative studies. One of the quantitative methods to assess usability is by using a questionnaire. The System Usability Scale (SUS) is one of the most widely used questionnaires to assess the usability of mobile apps [18]. It consists of 10 items which can be answered on a five-point Likert scale ranging from “strongly disagree” to “strongly agree”. SUS has been translated into various languages, i.e., Indonesian, Spanish, French, Dutch, Portuguese, Slovenian, Persian, German, and more recently into the Malay language, which is called the “*Skala Kebolehgunaan Aplikasi Mudah Alih”* (SKAMA) (Appendix A) [19]. However, to the best of our knowledge, there is no available questionnaire that assesses the usability of a self-management booklet. Thus, the first objective of this study was to adapt SKAMA into the EMPOWER-SUSTAIN Usability Questionnaire (E-SUQ) to assess usability of the EMPOWER-SUSTAIN Global CV Risks Self-Management Booklet^©^ exploring the extent to which this book can be used by patients with MetS in primary care and their personal experience. The second objective was to validate the questionnaire among patients with MetS in primary care.

## 2. Materials and Methods

### 2.1. Study Design and Population

This was a cross-sectional validation study that was conducted in three phases. The first phase consisted of adaptation and content validation of the E-SUQ. This was followed by face validation in the second phase, and field testing and psychometric evaluation of the questionnaire in the final phase. The study population included patients with MetS attending a university primary care clinic in Selangor, Malaysia. It was conducted based on the Consensus-based Standards for the selection of health Measurement Instruments (COSMIN) guideline and principles of good practice for validation of a questionnaire [20,21]. The conduct of the study is shown in the flowchart presented in Figure 1.

The inclusion criteria were as follows: (a) aged 18–80 years old; (b) attended the university primary care clinic for at least one year; (c) given the EMPOWER-SUSTAIN Global CV Risks Self-Management Booklet^©^ for ≥6 months; (d) had blood investigations (Fasting Plasma Glucose (FPG), Fasting Serum Lipid (FSL) and HbA1c) done in the last one year; (e) fulfilled at least 3 out of 5 diagnostic criteria for MetS based on the 2008 Joint Interim Statement definition [2] (i.e., waist circumference South Asian cut points: male ≥ 90 cm, female ≥ 80 cm; systolic blood pressure (BP) ≥ 130 and/or diastolic BP ≥ 85 mmHg or on treatment for hypertension (HTN); FPG ≥ 5.6 mmol/L or on treatment for elevated glucose; triglycerides (TG) ≥ 1.7 mmol/L or on treatment for dyslipidaemia; high-density lipoprotein-cholesterol (HDL-C): male < 1.0 mmol/L, female < 1.3 mmol/L or on treatment for dyslipidaemia); (f) were able to read and understand the Malay language and (g) willing to participate in the study.

The following patients were excluded from the study: (a) diagnosed with circulatory disorders requiring secondary care over the past year (e.g., acute coronary syndrome, stroke, transient ischemic attacks, peripheral vascular disease); (b) on renal dialysis; (c) presented with severe HTN (systolic BP > 180 mmHg and/or diastolic BP > 110 mmHg); (d) on radiotherapy/chemotherapy or palliative care; (e) had any form of mental disorder or cognitive impairment that would affect the ability to answer the questionnaire, for example, dementia or mental retardation; (f) pregnant; and (g) unable to give informed consent.

### 2.2. Study Tool

The E-SUQ in this study was adapted from SKAMA to assess the level of usability of the EMPOWER-SUSTAIN Global CV Risks Self-Management Booklet^©^. SKAMA is a 10-item questionnaire in the Malay language which assesses the usability of mobile apps, in which the response score is calculated using the 5-point Likert scale ranging from 1 (strongly disagree) to 5 (strongly agree). The item score on the positively phrased statements is subtracted by 1 (x − 1) and the item score on the negatively phrased statements is calculated by subtracting the score from 5 (5 − x) [22]. The overall score is computed as the summation of all item scores multiplied by 2.5, which gives the overall score that ranged from 0 (extremely poor usability) to 100 (excellent usability). The score value of > 68 is recommended by the original author to indicate the cut-off point for good usability of an app [23]. Permission to adapt and validate SKAMA into E-SUQ was obtained from the researchers.

### 2.3. Phase 1: Adaptation and Content Validation

The SKAMA underwent a process of adaptation into E-SUQ. Each item in the questionnaire which assessed the usability of mobile application was adapted to assess the usability of the self-management booklet. The word “mobile application” was substituted with “self-management booklet” in the Malay version. 

Content validation was conducted through an online survey by seven family medicine specialists who are experts in questionnaire validation. They are clinical experts with a special interest in patient empowerment and chronic disease management. According to the literature, content validation should include at least five experts to have sufficient control over chance agreement [24]. The original 10-item SKAMA was critically reviewed by the panels for clarity and relevance to the conceptual framework. The items were rated on a scale from 1 to 4 (1 = not relevant, 2 = somewhat relevant, 3 = quite relevant, 4 = highly relevant). The item level content validity index (I-CVI) was computed for each item by dichotomizing the 4-point scale. Items with a score of either 1 or 2 were recategorized into “not relevant” with 0 point. Items with a score of either 3 or 4 were recategorized into “relevant” with 1 point. The values (0 or 1) for each item were added up and then the total value was divided by the total number of experts [25]. An I-CVI value of at least 0.83 determined that the items were relevant and to be retained in the questionnaire [26]. Two rounds of content validity were conducted in this study because a major revision was required for item 5. After the second round, the scale level content validity index based on the average method (S-CVI/Ave) was computed to evaluate the relevance of the revised questionnaire. S-CVI/Ave is the average of the I-CVI scores for all items on the scale or the average of proportion relevance judged by all experts [25]. S-CVI/Ave value of >0.90 indicates that the item should be retained [27].

### 2.4. Phase 2: Face Validation

The face validation process was done through a face-to-face interview on a sample of 10 patients who met the inclusion and exclusion criteria. They were asked to comment on the questionnaire’s instructions, contents, terminology, comprehensibility and overall structure. Correction and fine-tuning of the E-SUQ were conducted by the research team based on the patient’s feedback. Face validity index (FVI) was used to evaluate the items in the form of clarity and comprehensibility of language and instructions used in the questionnaire [28]. The participants were requested to rate the comprehensibility of each item to the conceptual framework on a scale from 1 to 4 (1 = not understandable, 2 = somewhat understandable, 3 = understandable, 4 = very understandable). The item level-FVI (I-FVI) was computed for each item by dichotomizing the 4-point scale, with items scoring either 1 or 2 being recoded as 0 and items with a score of either 3 or 4 being recoded as 1 [29]. The values (0 or 1) for each item were added up and then the total value was divided by the total number of experts. Marzuki et al. suggested that if the number of raters is 10, the acceptable cut-off score of FVI is at least 0.83 [19]. This process produced the refined E-SUQ, which was ready to undergo the psychometric evaluation.

### 2.5. Phase 3: Field Testing and Psychometric Evaluation

In the final phase, the E-SUQ went through field testing for psychometric evaluation. The same inclusion and exclusion criteria were applied to the recruited participants. However, participants who were involved in Phase 2 were not included in Phase 3 of this study.

#### 2.5.1. Sample Size Determination

The sample size for the psychometric evaluation was calculated using the sample-to-variable ratio (SVR) of 20:1 [30]. The E-SUQ contains 10 items; therefore, a minimum sample size of 200 patients was required. After taking into consideration a 10% non-responder and non-eligibility rate, the study aimed to approach 220 patients.

#### 2.5.2. Sampling Method, Patient Recruitment and Data Collection

The EMPOWER-SUSTAIN Global CV Risks Self-Management Booklet^©^ was distributed to patients with MetS at the university primary care clinic from October 2019 to March 2020, to ensure that the patients had the booklet for at least 6 months prior to the usability data collection. 

Patients were recruited over four months from December 2020 to March 2021 when they attended the clinic for their follow-up appointment. The data collection was conducted by a trained research assistant to maintain a consistent method of collecting the data. Patients who were identified to have the booklet were approached in the nurse assessment room on the day of their follow-up appointment. They were briefed about the study and were invited to participate. A patient information sheet was given to participants to provide further details regarding the study, and their right for confidentiality was explained. Those who agreed to participate were screened for eligibility according to the inclusion and exclusion criteria. The patient’s eligibility was screened by reviewing the electronic medical record. A written informed consent was obtained from participants who agreed and were eligible to participate in the study. 

The usability data were collected using E-SUQ, which was distributed to participants to be self-administered. Clear verbal instructions were provided on how to complete the questionnaire without assistance. The participants were advised to seek clarification from the researcher for any questions. Participants took approximately 15 minutes to complete the questionnaire. Following completion, participants were asked to return the questionnaire to the researcher to be checked for completeness.

#### 2.5.3. Data Collection for Test-Retest

Thirty participants were recruited for the test-retest of E-SUQ. A date was given for them to come back to the clinic within 2–4 weeks to answer the same questionnaire. Participants were called one day before to remind them about the appointment. 

### 2.6. Statistical Analysis

Data entry and statistical analysis were performed using the latest IBM SPSS Statistics Program Version 25. During the data entry, the responses for negative statements (item 2, 4, 6, 8 and 10) were reversed. Data quality was examined using the percentage of missing data and mean score of E-SUQ with standard deviation (±SD) was calculated using the formula recommended by the SUS author [22]. Descriptive analysis was presented as frequency and percentage for categorical data. Normally distributed continuous data were expressed as mean with ±SD and non-normally distributed data were expressed as median with interquartile range (IQR). 

Psychometric elements of E-SUQ were examined in three parts. Firstly, the factorability of the 10 items was examined to determine the suitability of the data to undergo factor analysis. The sampling adequacy was assessed using the Kaiser-Meyer-Olkin (KMO), whereas the appropriateness of data was conducted using Bartlett’s test of sphericity. The data are considered to be suitable for factor analysis when the KMO value is >0.50 [31] and if the p-value of Bartlett’s test of sphericity is <0.05 [32]. 

Secondly, factor extraction using principal component analysis (PCA) was conducted to identify the dimensionality of the 10 items of E-SUQ. The number of factors to retain was determined using the following tests: the rule of eigenvalue >1 according to Kaiser’s Criteria, >50% cumulative percentage of variance and the scree plot [33]. The retained factors were then rotated using varimax rotation with the factor loading significance set at >0.4 [34].

Thirdly, the internal consistency and test-retest reliability analyses were conducted to determine the reliability of E-SUQ. A Cronbach α coefficient and corrected item-total correlation (CITC) were used to measure internal consistency. A minimum value of Cronbach’s α coefficient of 0.7 [35] and a minimum CITC range (r) of 0.3 were considered as reliable [36]. The test-retest reliability was examined using the intra-class correlation coefficient (ICC) to assess the temporal stability of the item. The ICC estimates and their 95% confidence intervals were calculated based on the mean rating (k = 30), absolute agreement and two-way mixed-effect model (ICC [2,k]). The ICC values <0.5 indicate poor reproducibility, between 0.5 and 0.75 indicate moderate reproducibility, between 0.75 and 0.9 indicate good reproducibility, and values >0.90 indicate excellent reproducibility [37].

## 3. Results

### 3.1. Phase 1: Content Validation of the E-SUQ 

Table 1 shows the calculated I-CVI for the 10 items of E-SUQ. The I-CVI were found to be between 0.85 and 1.0 after two rounds of content validity analysis. In the first round, the I-CVI score for item 5 was 0.57, in which three experts found that the statement was not clear and relevant to the construct being measured. According to the three experts who gave low scores for item 5, they found that the sentence was unclear as there was no example given for the content of self-management booklet such as *My Profile*, *My Cardiovascular Risks*, *My Treatment Targets*, *My Check-Up*, *My Weight Management*, *My Smoking Habit*, *My Self-Management*, and *My Medication* sections. Therefore, item 5 was revised and examples were added to make the item clearer and more relevant. The three expert panels who scored low for item 5 were invited to rate it again in the second round of content validation. The I-CVI for item 5 after the second round was 1.0. The calculated S-CVI/Ave for the 10 items after the second round was 0.98. Therefore, all 10 items were retained.

### 3.2. Phase 2: Face Validation of the E-SUQ 

The face validation was performed on 10 participants. Five participants gave suggestions to improve the wording of item 7 for better understanding and clarity. They suggested that item 7, which began with “I would imagine that” (“*Saya membayangkan*”), was changed to “I think that” (“*Saya berpendapat*”). Therefore, item 7 of the questionnaire was revised as suggested by the participants. This amendment is consistent with the other items, which encourage participants to express their personal views through opening statements such as “ I think that” or “I find that”. I-FVI was calculated and the results are shown in Table 2. The calculated I-FVI ranged between 0.9 and 1.0 for each item, which indicates an acceptable I-FVI level. The refined ESUQ was ready to undergo psychometric evaluation (Appendix B).

### 3.3. Phase 3: Psychometric Evaluation of E-SUQ

#### 3.3.1. Recruitment for Field Testing

The EMPOWER-SUSTAIN Global CV Risks Self-Management Booklet^©^ was distributed to 677 patients with MetS from October 2019 to March 2020. Patient recruitment for the usability data collection was done 6 months later in December 2020 to March 2021. In total, 210 patients were approached and invited to participate in the study when they attended the clinic for their follow-up appointment. Out of these, 205 patients (97.6%) fulfilled the eligibility criteria and agreed to participate. A total of 205 participants were recruited and completed the E-SUQ.

#### 3.3.2. Demographic and Clinical Characteristics of Participants

Table 3 summarizes the demographic and clinical characteristics of the participants in this study. The mean age was 63.5 (SD ± 7.5). The age of the participants ranged between 42 and 79 years old. Out of 205 participants, the majority were males (53.2%), Malay (88.3%), married (85.4%), and had secondary and tertiary educational level (93.7%), retiree (60.5%), came from the B40 income group (52.2%), obese (62%), never smoke (72.7%) and reported to have good health status (87.8%).

#### 3.3.3. Psychometric Properties

A total of 205 responses were included in the psychometric evaluation. There was no missing response to the questionnaire items, indicating good quality of the data. The means score of E-SUQ was 77.3 (SD ± 13.8). In the correlation matrix, all of the items had a correlation of ≥0.30 with at least one other item, indicating reasonable factorability. The sample size for this study was adequate, as evidenced by the high KMO value of 0.871 and Bartlett’s test of sphericity *p*-value of <0.05. These two values emphasized the appropriateness for factor analysis. 

All items of the E-SUQ underwent PCA with subsequent varimax rotations. The analysis extracted two factors with eigenvalues >1 according to the Kaiser’s Criteria. The two extracted factors explained 60.6% of the cumulative percentage of variance. The elbow of the scree plot occurred between the second and third component, suggesting two factors to be retained. 

Table 4 shows the loading values based on the PCA and varimax rotation analysis. The loading values were above the minimum cut-off point value of 0.40, ranging between 0.484 and 0.804. The highest and lowest communalities were for item 7 with a value of 0.690 and item 2 with a value of 0.383, respectively. This means that item 7 accounted for 69.0% of its variability. All the 10 items of E-SUQ were valid and can be used to measure usability of the self-management booklet. Items number 1, 3, 5, 7 and 9 are positively phrased statements loaded onto Factor 1. Items number 2, 4, 6, 8 and 10 are negatively phrased statements loaded onto Factor 2. Therefore, Factor 1 is labeled as “Positive Tone” and Factor 2 is labeled as “Negative Tone”. These findings indicate that E-SUQ is characterised by the Positive/Negative Tone model. 

In addition, the reliability analysis was conducted to assess the internal consistency of the items in E-SUQ. Table 5 shows the results of the analysis. The calculated CITC was between 0.30 and 0.52. The overall Cronbach’s α coefficient for E-SUQ was 0.77. The Cronbach’s α coefficient for Factor 1 was 0.88 and for Factor 2 was 0.75. All the values met the minimum cut-off of 0.7, indicating that E-SUQ was reliable.

Table 6 shows the test-retest reliability results among 30 participants. The demographic characteristics of the 30 participants were similar to the other 175 participants who were not involved in the test-retest reliability study (Appendix C). The overall ICC value was 0.85, indicating that ESUQ was stable over time. The individual item values ranged between 0.58 and 0.88, indicating good reproducibility.

## 4. Discussion

To the best of our knowledge, there is no published study on the validation of a usability questionnaire to assess a self-management booklet. Our study is the first to demonstrate that E-SUQ is a valid and reliable tool to assess the usability of the EMPOWER-SUSTAIN Global CV Risks Self-Management Booklet^©^. E-SUQ also underwent a thorough adaptation and validation process based on the recommended guidelines [20].

The data set of our study is of good quality as there were no missing values. The mean score of E-SUQ among patients with MetS was 77.3 (SD ± 13.8). Direct comparison with other studies assessing usability of the self-management booklet could not be made as there was no published literature in this area. Our findings can be compared with usability of mobile apps as there is numerous published evidence in this area. The mean score in our study is comparable to the mean score of SKAMA (72.9 (SD ± 11.5)) which was used to assess a colorectal community education mobile app in Malaysia [38]. Other validation studies that used the original SUS had reported higher mean scores. These include studies among patients with type 2 diabetes mellitus in Utah which showed a mean score of 80.5 (SD ± 11.5) [39] and among patients living with chronic pain in Norway which showed mean score of 85.7 (SD ± 12.9) [40]. In contrast, a usability study of a cognitive behavioural therapy app using SUS among mental healthcare providers in six European countries yielded a lower mean score of 67.9 (SD ± 18.8) [41]. This might be due to comprehension problems as there was no translation of SUS to the local language in these European countries. 

Content validation was conducted twice in our study due to the need to improve clarity and relevance for item 5. Several studies supported multiple iterations in the process of content validation to improve the clarity and relevance of the items representing the underlying construct [26,42,43]. With regards to face validity, ESUQ was found to be clear and comprehensible with I-FVI ranged between 0.9 and 1.0. Our finding is comparable to the I-FVI of SKAMA which ranged between 0.8 and 1.0 [19]. 

Factor analysis in this study revealed a two-factor solution in which items 1, 3, 5, 7 and 9 loaded onto Factor 1 (“Positive Tone”) and items 2, 4, 6, 8 and 10 loaded onto Factor 2 (“Negative Tone”). This is known as the tone model, which has been described in various other studies [41,44,45,46]. However, two studies did not support the tone model as it was of limited theoretical interest in relation to usability [44,45]. They recommended to interpret SUS as a unidimensional measure of perceived usability [44,45]. A more recent study using confirmatory factor analysis (CFA) supported a two-factor model of “Usability” and “Learnability” as the best fitted model compared to the unidimensional or the tone model [41].

The overall Cronbach’s α coefficient found in our study was 0.77, which indicates that E-SUQ is reliable to assess the usability of a self-management booklet. This finding is comparable to SKAMA, which has a Cronbach α of 0.85 [19]. Likewise, our finding is also comparable to the reliability of the SUS in other studies which showed Cronbach’s α of 0.91 [41] and 0.88 [46].

In terms of test-retest reliability, our study demonstrates an overall value of 0.85, indicating good reproducibility. This finding is comparable to the Persian version of SUS, where the value of overall ICC was 0.96 [47]. In contrast, the Portuguese version of SUS showed a weak overall ICC value of 0.36 [48].

### 4.1. Strength and Limitation

The strengths of this study include having a good quality data set as there was no missing value and the mean score showed good usability at 77.3 (SD ± 13.8). Another strength of this study was the high response rate of 97.6%. Our study is the first to demonstrate that E-SUQ is a valid and reliable tool to assess the usability of a self-management booklet. To date, there is no other published questionnaire available that can be regarded as an “anchor tool” or “gold standard”. Therefore, convergent validity could not be performed. There were also limitations in this study. Firstly, the population was from a university primary care clinic with Malay ethnicity (88.3%) constituting the vast majority of participants. As a result, the outcome of this validation study may not be generalizable to other primary care clinics in Malaysia with multiracial populations such as Chinese and Indian. Secondly, patients with MetS who were given the EMPOWER-SUSTAIN Global CV Risks Self-Management Booklet^©^ were chosen conveniently. This may introduce sampling bias. To minimize this, all patients who were identified to have the booklet on their appointment day were invited to participate in the study. Thirdly, CFA could not be conducted to confirm the dimensionality of the item as it requires a larger sample size of at least 300. This was not feasible considering the limited time frame given to complete this study. The COVID-19 pandemic also had an impact on the number of patients coming to our clinic, resulting in a longer time taken to collect the data. Lastly, the validated E-SUQ questionnaire can only be utilized by those who can read and understand the Malay language.

### 4.2. Implication for Further Research and Clinical Practice

This study has proven that E-SUQ is a valid and reliable questionnaire to assess the level of usability of the EMPOWER-SUSTAIN Global CV Risks Self-Management Booklet^©^ among patients with MetS attending a university primary care clinic. Further validation study involving patients from other primary care clinics in Malaysia is recommended to improve its generalizability. ESUQ should also be translated into Mandarin and Tamil so that the level of usability can be assessed among other major ethnic groups in Malaysia such as Chinese and Indian. As there were many controversies with the factor structure of the original SUS, future studies using CFA should be performed to confirm the dimensionality of E-SUQ. Further research to determine the factors associated with usability of the booklet is also recommended to support its widespread use among patients with MetS.

## 5. Conclusions

E-SUQ is a valid and reliable instrument to assess the level of usability of a self-management booklet among patients with MetS in primary care. The psychometric properties of the final 10-item E-SUQ show that it has good construct validity, high internal consistency and good test-retest reproducibility. Establishing usability of a self-management booklet will support its widespread use which, in turn, would enhance patient’s empowerment and improve their health outcomes.

## Figures and Tables

**Figure 1 ijerph-18-09405-f001:**
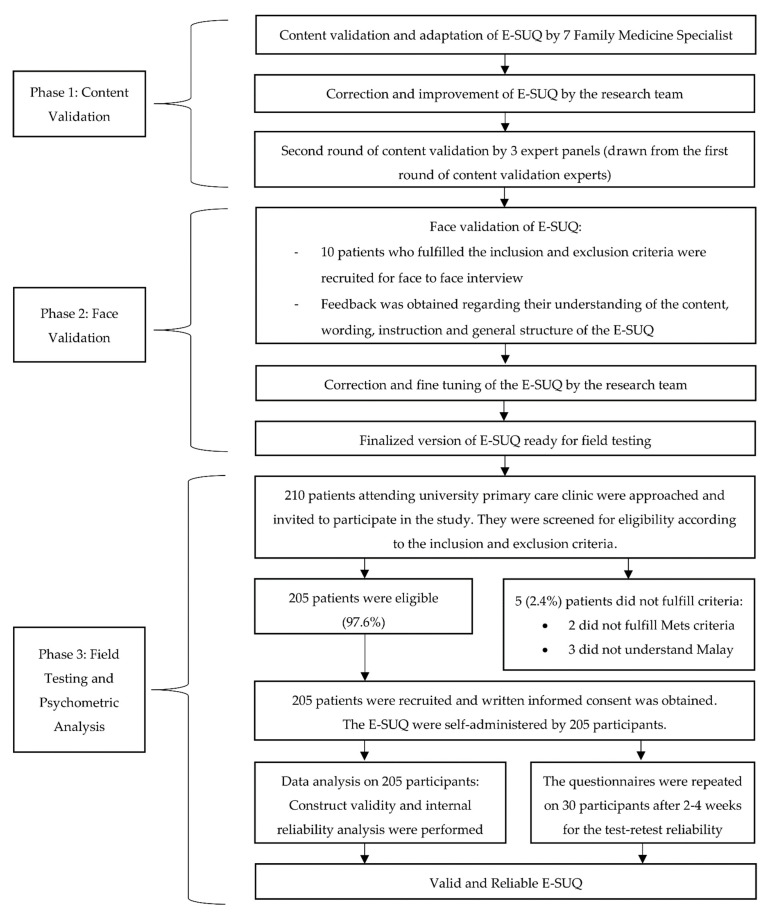
Flowchart of the study.

**Table 1 ijerph-18-09405-t001:** Final round of content validity index.

Item	E ^a^ 1	E2	E3	E4	E5	E6	E7	Experts in Agreement	I-CVI ^b^
Q1	3	3	4	4	4	3	4	7	1.0
Q2	4	4	4	4	4	4	4	7	1.0
Q3	4	4	4	4	4	4	4	7	1.0
Q4	4	4	4	4	4	4	4	7	1.0
Q5	3	3	4	4	4	3	3	7	1.0
Q6	4	4	4	4	4	2	4	6	0.85
Q7	3	3	3	4	4	4	4	7	1.0
Q8	4	4	4	4	4	4	4	7	1.0
Q9	4	4	4	4	4	3	4	7	1.0
Q10	4	4	4	4	4	4	4	7	1.0
**Content validity index average (S-CVI/Ave)**									0.98

^a^ E: Expert; ^b^ I-CVI: Item content validity index.

**Table 2 ijerph-18-09405-t002:** Face validity index.

Item	R ^a^ 1	R2	R3	R4	R5	R6	R7	R8	R9	R10	Raters in Agreement	I-FVI ^b^
Q1	4	3	4	4	4	4	3	4	3	4	10	1.0
Q2	4	4	4	4	4	4	4	3	4	4	10	1.0
Q3	4	4	4	4	4	4	4	4	4	4	10	1.0
Q4	4	4	4	4	4	4	4	4	4	4	10	1.0
Q5	4	4	4	4	4	4	4	4	4	4	10	1.0
Q6	3	4	4	4	4	4	4	2	4	4	9	0.9
Q7	3	4	4	4	4	4	4	4	4	4	10	1.0
Q8	4	4	4	4	4	4	4	3	3	4	10	1.0
Q9	4	4	4	4	4	3	4	4	4	4	10	1.0
Q10	4	3	3	4	4	2	4	4	4	3	10	0.9

^a^ R: Rater ^b^ I-FVI: Item face validity index.

**Table 3 ijerph-18-09405-t003:** Demographics and clinical characteristics of participants (n = 205).

Characteristics of Participants	Frequency, n (%)	Mean (±SD)
**Age** (years)		63.5 (±7.5)
**Gender**		
Male	109 (53.2)	
Female	96 (46.8)	
**Ethnicity**		
Malay	181 (88.3)	
Chinese	15 (17.3)	
Indian	7 (3.4)	
Others	2 (1.0)	
**Marital Status**		
Single	4 (2.0)	
Married	175 (85.4)	
Widow/Widower	22 (10.7)	
Divorced	4 (2.0)	
**Educational Level**		
No formal education	1 (0.5)	
Primary	12 (5.9)	
Secondary	93 (45.4)	
Tertiary	99 (48.3)	
**Occupation**		
Unemployed	40 (19.5)	
Employed	41 (20.0)	
Retiree	124 (60.5)	
**Household income per month ***		
B40 (<RM 4360)	107 (52.2)	
M40 (RM 4360–9619)	71 (34.6)	
T20 (>RM 9619)	27 (13.2)	
**Body Mass Index, kg/m²**		
Underweight (<18.5)	0 (0.0)	
Normal (18.5–22.9)	12 (5.9)	
Overweight (23.0–27.4)	66 (32.1)	
Obese (≥27.5)	127 (62.0)	
**Smoking status**		
Never smoke	149 (72.7)	
Active smoker	23 (11.2)	
Ex-smoker	33 (16.1)	
**Self-report health status**		
Excellent	2 (1.0)	
Very Good	10 (4.9)	
Good	180 (87.8)	
Fair	13 (6.3)	

* Based on Report of Household Income and Basic Amenities Survey 2016 by Department of Statistics, Malaysia.

**Table 4 ijerph-18-09405-t004:** Factor loading.

Item	Question	Factor 1	
		1	2
ESUQ1	I think that I would like to use this self-management booklet frequently.	0.804	
ESUQ2	I found the self-management booklet unnecessarily complex.		0.484
ESUQ3	I thought the self-management booklet was easy to use.	0.796	
ESUQ4	I think I would need the support of a technical person to be able to use the self-management booklet.		0.662
ESUQ5	I found that the contents of the self-management booklet were related to each other (Example: Cardiovascular risk factors → Blood test results → Weight management → Smoking habit → Home blood pressure monitoring/Self blood sugar monitoring → List of medication).	0.803	
ESUQ6	I found that there was inconsistency in the content of self-management booklet.		0.734
ESUQ7	I think the self-management booklet is easy to learn.	0797	
ESUQ8	I found the self-management booklet is difficult to use.		0.695
ESUQ9	I felt confident to use the self-management booklet.	0.794	
ESUQ10	I needed to learn a lot of things before I could get going with the self-management booklet.		0.779

**Table 5 ijerph-18-09405-t005:** Internal consistency.

Item	Scale Mean if Item Deleted	Scale Variance if Item Deleted	Corrected Item-Total Correlation	Squared Multiple Correlation	Cronbach’s Alpha if Item Deleted
ESUQ 1	39.37	8.38	0.49	0.80	0.75
ESUQ 2	39.30	8.91	0.30	0.43	0.77
ESUQ 3	39.57	8.05	0.51	0.62	0.75
ESUQ 4	39.57	8.25	0.52	0.60	0.75
ESUQ 5	39.37	8.31	0.52	0.83	0.75
ESUQ 6	39.50	8.47	0.35	0.57	0.77
ESUQ 7	39.53	8.33	0.41	0.53	0.76
ESUQ 8	39.47	8.05	0.47	0.70	0.75
ESUQ 9	39.40	8.25	0.46	0.52	0.75
ESUQ 10	39.43	8.67	0.40	0.55	0.76
**Overall Cronbach’s alpha value**					**0.77**

**Table 6 ijerph-18-09405-t006:** Test-retest reliability intra-class correlation coefficient (n = 30).

Item	Question	ICC [2,k] (95% CI)
ESUQ 1	I think that I would like to use this self-management booklet frequently.	0.80 (0.57–0.90)
ESUQ 2	I found the self-management booklet unnecessarily complex.	0.66 (0.29–0.84)
ESUQ 3	I thought the self-management booklet was easy to use.	0.58 (0.11–0.80)
ESUQ 4	I think I would need the support of a technical person to be able to use the self-management booklet.	0.73 (0.43–0.87)
ESUQ 5	I found that the contents of the self-management booklet were related to each other (Example: Cardiovascular risk factors → Blood test results → Weight management → Smoking habit → Home blood pressure monitoring/Self blood sugar monitoring → List of medication)	0.76 (0.49–0.88)
ESUQ 6	I found that there was inconsistency in the content of the self-management booklet.	0.71 (0.40–0.86)
ESUQ 7	I think the self-management booklet is easy to learn.	0.88 (0.75–0.94)
ESUQ 8	I found the self-management booklet is difficult to use.	0.80 (0.58–0.90)
ESUQ 9	I felt confident to use the self-management booklet.	0.57 (0.09–0.80)
ESUQ 10	I needed to learn a lot of things before I could get going with the self-management booklet.	0.67 (0.31–0.84)
	**Overall value**	**0.85**

## Data Availability

Data are kept at the Department of Primary Care Medicine, Universiti Teknologi MARA, in Selangor, Malaysia. Data will be shared upon request and they are subjected to the data protection regulations.

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
