# Peer review of "Adaptation and Psychometric Validation of the EMPOWER-SUSTAIN Usability Questionnaire (E-SUQ) among Patients with Metabolic Syndrome in Primary Care"

_ijerph, 2021, doi:10.3390/ijerph18179405_

Round 1

Reviewer 1 Report

this is a paper on the development and validation of an usability scale specific to metabolic syndrome patient material

Major issues

1 in the introductory part it is not discussed on the practical usefulness of this scale : is it to improve the content? is it to understand the perception of the patient? is it to monitor some intervetions?

2 the validation methodology although very elaborated and well described does not does not include an "anchor tool" which your scale should be compared against. if so please include . if you deliberately chose not to please argument

3 a conclusion based on the psychometric results obtained is missing ( ie when to use it which are the strong points and which are the weaknesses

Author Response

Response to Reviewer 1 Comments
Point 1: In the introductory part it is not discussed on the practical usefulness of this scale: is it to improve the content? Is it to understand the perception of the patient? Is it to monitor some interventions?
Response 1: Thank you for your comment.
We have explained in the introductory part that the E-SUQ scale was to assess the usability of the EMPOWER-SUSTAIN Global Cardiovascular Risks Self-Management Booklet as mentioned in page 2, line 80-84. Based on your comment, we have revised the sentence to make it clearer in describing the usefulness of the questionnaire in the study.
“Thus, the first objective of this study was to adapt SKAMA into the EMPOWER-SUSTAIN Usability Questionnaire (E-SUQ) to assess usability of the EMPOWER-SUSTAIN Global CV Risks Self-Management Booklet© exploring the extent to which this book can be used by patients with MetS in primary care and their personal experience. The second objective was to validate the questionnaire among patients with MetS in primary care.”

Point 2: The validation methodology although very elaborated and well described does not include an "anchor tool" which your scale should be compared against. If so please include. If you deliberately choose not to, please provide argument.
Response 2: Thank you for your comment. “Our study is the first to demonstrate that E-SUQ is a valid and reliable tool to assess the usability of a self-management booklet. To date, there is no other published questionnaire available that can be regarded as an ‘anchor tool’ or ‘gold standard’. Therefore, convergent validity could not be performed.” We have added this argument in the Strength and Limitation section.

Point 3: A conclusion based on the psychometric results obtained is missing (i.e. when to use it which are the strong points and which are the weaknesses)
Response 3: Thank you for your comment. We have added the following statement in the Conclusion:
“The psychometric properties of the final 10-item E-SUQ shows it has good construct validity, high internal consistency and good test-retest reproducibility.”

Reviewer 2 Report

In this paper, the authors adapted SUS to E_SUQ to determine its validity and reliability in assessing usability of a self-management booklet. It is an interesting topic. The following are a few questions and concerns I have regarding the paper. 

  1. Why there are only 3 experts in the second round of content validation? How these 3 experts selected from the 7?
  2. 43.4% of Malaysian adults have MetS. The booklet was distributed to patients with MetS at the university primary care clinic from October 2019 to 221 March 2020 and patients were recruited over three months from December 2020 to March 2021. I think there should be more than 210 patients with MetS visit the clinic within this period. How were these 210 patients selected?
  3. 30 participants were selected for test-retest. How these participants were selected? There should be a table like Table 3 to show the demographics of the 30 participants and whether they are different from the others. There should also be a table comparing the answers of the 30 participants to the others at the first time of the test. Are the 30 participants able to represent all participants?
  4. Tables 4 and 6 should be an English translated version. The authors could put the original Malay version in the supplementary.
  5. We usually say CVD (cardiovascular disease) risk instead of CV risk. 
  6. The reference 31 should be "Kaiser, Henry F., and John Rice. "Little jiffy, mark IV." Educational and psychological measurement 34.1 (1974): 111-117."

Author Response

Response to Reviewer 2 Comments
Point 1: Why there are only 3 experts in the second round of content validation? How were these 3 experts selected from the 7?
Response 1: Thank you for your comment. Based on the published literature by Polit et al., (2007), a smaller group of experts (perhaps 3-5) can be used to evaluate the relevance of revised item in the second round of content validity. The rater can be drawn from the same pool of experts as in the first round. Using a subset of experts from the first round has distinct advantage, because information from the first round can be used to select the most capable judges. Qualitative feedback from an expert in round 1 in the form of useful comments about the items indicates content capability and a strong commitment to the project.
Based on this recommendation, the three experts who gave low scores for item 5 were selected to rate the item again in the second round of content validation as mentioned in page 7, line 288-296 as follows:
“In the first round, the I-CVI score for item 5 was 0.57 in which three experts found that the statement was not clear and relevant to the construct being measured. According to the three experts who gave low scores for item 5, they found the sentence was unclear as there was no example given for the content of self-management booklet such as My Profile, My Cardiovascular Risks, My Treatment Targets, My Check-Up, My Weight Management, My Smoking Habit, My Self-Management, and My Medication sections. Therefore, item 5 was revised and examples were added to make the item clearer and more relevant. The three expert panels who scored low for item 5 were invited to rate it again in the second round of content validation.”

Point 2: 43.4% of Malaysian adults have MetS. The booklet was distributed to patients with MetS at the university primary care clinic from October 2019 to 21 March 2020 and patients were recruited over three months from December 2020 to March 2021. I think there should be more than 210 patients with MetS visit the clinic within this period. How were these 210 patients selected?
Response 2: Thank you for your comment. The booklet was successfully distributed to a total of 677 patients between October 2019 to March 2020. Although the total number of patients with MetS in our clinic is estimated to be around 2000 patients, the reasons why we only managed to distribute to 677 patients from Dec 2020 to March 2021 are as follows:
1. Due to the current pandemic Covid-19 situation, the number of patients attending our clinic was reduced.
2. The number of appointments per day was also limited to reduce the congestion in the clinic.
3. Some of the patients had their follow-up appointments done online through a Virtual Clinic.
According to our sample size calculation, a minimum sample size of 200 patients was required to achieve the study objectives. The selection method of the 210 patients included in this study is described in the Method section - page 6, line 228-238, as follow:
“Patients who were identified to have the booklet were approached in the nurse assessment room on the day of their follow-up appointment. They were briefed about the study and were invited to participate. A patient information sheet was given to participants to provide further details regarding the study and their right for confidentiality was explained. Those who agreed to participate were screened for eligibility according to the inclusion and exclusion criteria. The patient’s eligibility was screened by reviewing the electronic medical record. A written informed consent was obtained from participants who agreed and were eligible to participate in the study.”
We have also added the following in the Result section 3.3.1 Recruitment for field testing: “The EMPOWER-SUSTAIN Global CV Risks Self-Management Booklet© was distributed to 677 patients with MetS from October 2019 to March 2020. Patient recruitment for the usability data collection was done 6 months later in December 2020 to March 2021. In total, 210 patients were approached and invited to participate in the study when they attended the clinic for their follow-up appointment. Out of these, 205 patients (97.6%) fulfilled the eligibility criteria and agreed to participate. A total of 205 participants were recruited and completed the E-SUQ.”

Point 3: 30 participants were selected for test-retest. How were these participants selected? There should be a table like Table 3 to show the demographics of the 30 participants and whether they are different from the others. There should also be a table comparing the answers of the 30 participants to the others at the first time of the test. Are the 30 participants able to represent all participants?
Response 3: Thank you for your comment. Based on the recommendation by Bujang and Baharum 2017, the number of participants for the test-retest analysis is 30. The 30 patients selected for the test-retest were invited from the same pool of 205 patients that participated for the psychometric evaluation. Therefore, the demographics of the 30 participants have been described in Table 3 together with the demographics of the study population for the field testing (n=205).

Point 4: Tables 4 and 6 should be an English translated version. The authors could put the original Malay version in the supplementary.
Response 4: Thank you for your suggestion. We have revised Tables 4 and 6 and added the English version of the scale.

Point 5: We usually say CVD (cardiovascular disease) risk instead of CV risk.
Response 5: Thank you for your suggestion. The word cardiovascular (CV) in this study originated from the name of the booklet which is the EMPOWER-SUSTAIN Global Cardiovascular Risks Self-Management Booklet. Therefore, we have decided to use the same abbreviation for the cardiovascular risk (CV risk).

Point 6: The reference 31 should be "Kaiser, Henry F., and John Rice. "Little jiffy, mark IV." Educational and psychological measurement 34.1 (1974): 111-117."
Response 6: Thank you for your suggestion. We have revised the reference 31 as recommended.

Reviewer 3 Report

Manuscript ID: ijerph-1315552

Manuscript title: Adaptation and Psychometric Validation of the EMPOW- 3 ER-SUSTAIN Usability Questionnaire (E-SUQ) among Patients 4 with Metabolic Syndrome in Primary Care

This study adapted the System Usability Scale (or ‘Skala Kebolehgunaan Aplikasi Mudah Alih’, SKAMA) into the EMPOWER-SUSTAIN Usability Questionnaire (E-SUQ) to assess usability of the EMPOWER-SUSTAIN Global CV Risks Self-Management Booklet© and validated the questionnaire among patients with metabolic syndrome in a primary care setting. The research is very well conducted and the reporting very written. Main guidelines for this study design and reporting were followed. Statistical analysis seems properly applied, with minor suggestions. Discussion is limited to the new findings and the conclusions are supported by the results.

Minor comments

  1. Statistical analysis (abstract and main text). Please fully specify the ICC[k,q] model for the reported values.

  1. Statistical analysis. Consider calculating the standard error of measurement (SEM) as it is also informative about the validity of the new instrument.

  1. Sample size calculation. Just a note for future studies. If you approach the 220 patients and experience the expected non-respondent rate of exactly 10%, you will end up with (220-22=) 198 participants. You would have to approach 222 participants to allow 22 (10%) participants as non-respondents and end up with the minimum 200 as required.

Author Response

Response to Reviewer 3 Comments
Point 1: Statistical analysis (abstract and main text). Please fully specify the ICC[k,q] model for the reported values.
Response 1: Thank you for your comment. We have specified the ICC [k,q] model as recommended and revised it to ICC [2,k] model in both the abstract and main text.

Point 2: Statistical analysis. Consider calculating the standard error of measurement (SEM) as it is also informative about the validity of the new instrument.
Response 2: Thank you for your suggestion. In this study, we determined the test-retest reliability of the scale by assessing the intraclass correlation coefficients with 95% CI. Our study shows that the test-retest reliability of this scale indicates good reproducibility.
The standard error of measurement (SEM) assesses the measurement error indicating how much two scores can vary in stable patients. It provides additional information about observed measurement error in the test-retest reliability of a scale. However, most published literatures on test-retest reliability of a scale do not provide this extra information. Therefore, for this study, we feel it is sufficient to provide the intraclass correlation coefficients with 95% CI, without the SEM.

Point 3: Sample size calculation. Just a note for future studies. If you approach the 220 patients and experience the expected non-respondent rate of exactly 10%, you will end up with (220-22=) 198 participants. You would have to approach 222 participants to allow 22 (10%) participants as non-respondents and end up with the minimum 200 as required.
Response 3: Thank you for your suggestion. We will take note of the recommendation for our future studies.

Round 2

Reviewer 1 Report

The authors did not go into more details but did some improvements

Author Response

Thank you for your comments

Reviewer 2 Report

The authors have addressed all my comments except the following one. 

I know the 30 patients selected for the test-retest have been described in Table 3. The issue is the authors should show that these 30 patients were randomly selected from the 205 patients. It would be bias if the 30 patients were all male or all Malay. It could be also bias if the answers of the 30 patients are different from the other 175 patients. Maybe the 30 patients gave high scores to all questions. That is why the authors need to show that the demographics and answers of the 30 patients are similar to the other 175 patients. 

Author Response

Point 1:
I know the 30 patients selected for the test-retest have been described in Table 3. The issue is the authors should show that these 30 patients were randomly selected from the 205 patients. It would be bias if the 30 patients were all male or all Malay. It could be also bias if the answers of the 30 patients are different from the other 175 patients. Maybe the 30 patients gave high scores to all questions. That is why the authors need to show that the demographics and answers of the 30 patients are similar to the other 175 patients.

Response 1: Thank you for your comment. The demographic characteristics of the 30 participants were similar to the other 175
participants who were not involved in the test-retest reliability study. The mean score of E-
SUQ among 30 participants was 77.1 (SD ± 21.2) and this is comparable to the mean score of
175 participants who were not involved, 77.5 (SD ± 14.1). This is shown in Appendix C.